# Numerical analysis of the impact of cytoskeletal actin filament density alterations onto the diffusive vesicle-mediated cell transport

**Daniel Ch. Haspinger**[1], **Sandra Klinge**[2], **Gerhard A. Holzapfel**[1,3]*

**1** Institute of Biomechanics, Graz University of Technology, Graz, Austria, **2** Chair of Structural Mechanics and Analysis, TU Berlin, Berlin, Germany, **3** Faculty of Engineering Science and Technology, Norwegian University of Science and Technology (NTNU), Trondheim, Norway

* holzapfel@tugraz.at

**Data Availability Statement:** All relevant data are within the manuscript and its Supporting information files.

## Abstract

The interior of a eukaryotic cell is a highly complex composite material which consists of water, structural scaffoldings, organelles, and various biomolecular solutes. All these components serve as obstacles that impede the motion of vesicles. Hence, it is hypothesized that any alteration of the cytoskeletal network may directly impact or even disrupt the vesicle transport. A disruption of the vesicle-mediated cell transport is thought to contribute to several severe diseases and disorders, such as diabetes, Parkinson's and Alzheimer's disease, emphasizing the clinical relevance. To address the outlined objective, a multiscale finite element model of the diffusive vesicle transport is proposed on the basis of the concept of homogenization, owed to the complexity of the cytoskeletal network. In order to study the microscopic effects of specific nanoscopic actin filament network alterations onto the vesicle transport, a parametrized three-dimensional geometrical model of the actin filament network was generated on the basis of experimentally observed filament densities and network geometries in an adenocarcinomic human alveolar basal epithelial cell. Numerical analyzes of the obtained effective diffusion properties within two-dimensional sampling domains of the whole cell model revealed that the computed homogenized diffusion coefficients can be predicted statistically accurate by a simple two-parameter power law as soon as the inaccessible area fraction, due to the obstacle geometries and the finite size of the vesicles, is known. This relationship, in turn, leads to a massive reduction in computation time and allows to study the impact of a variety of different cytoskeletal alterations onto the vesicle transport. Hence, the numerical simulations predicted a 35% increase in transport time due to a uniformly distributed four-fold increase of the total filament amount. On the other hand, a hypothetically reduced expression of filament cross-linking proteins led to sparser filament networks and, thus, a speed up of the vesicle transport.

**Funding:** This research was supported by the German Research Foundation (DFG), number KL2678/7-1, to SK and by the Austrian Science Fund (FWF), number 3431-N32 to GAH. The funders had no role in study design, data collection and analysis, decision to publish, or preparation of the manuscript.

**Competing interests:** The authors have declared that no competing interests exist.

## Author summary

Many vital processes in our eukaryotic cells and organs require an astonishingly precise routing of intermediate products to various intra- and extracellular destinations using vesicles as transporters. This can be illustrated by numerous examples, such as the production and destruction of proteins, the export of neurotransmitters or insulin to the extracellular domain, etc. However, the inside of a cell is tightly packed with numerous structural scaffoldings (filaments), which serve as obstacles and impede the vesicle motion. It is thought that any disturbances of the vesicle-mediated cell transport contribute to numerous degenerative diseases and disorders, which highlights the clinical relevance for investigating this intracellular transport mechanism by developing computational models and performing experimental studies. In this study, we numerically quantified how different specific alterations of the filament density inside a human lung cell—due to changed mechanical loadings or genetic disorders of proteins being responsible for filament branching—affect the diffusion of vesicles inside the intracellular fluid. Therefore, based on the concept of homogenization, a computationally efficient numerical method was developed and utilized to simulate the diffusion of vesicles inside the whole cell, considering the detailed structural information of the filament network.

## Introduction

One of the most characteristic property of a eukaryotic cell is the utterly high level of organization of its internal transport processes, in particular the vesicle-mediated cell transport. It is a biological process where, e.g., soluble proteins either enter the cell or get released into the extracellular matrix, known as endo- or exocytosis. In case of the endocytosis, the selected cargo together with some plasma membrane lipids and extracellular fluid become fully internalized into the cell forming a spherical vesicle. Diameters between 20 and 100 nm for intracellular transport vesicles have been reported [1–5]. The budding and scission process represents the *de novo* production of internal membranes from the plasma membrane lipid bilayer (donor membrane). Afterwards, the vesicles either diffuse passively through the cytosol (slow transport) or move actively along microtubules (fast transport) by means of motor proteins, such as kinesin or dynein. The journey continues until the vesicle finds the correct destination (tethering), docks to and fuses with the acceptor membrane, where it finally delivers the cargo. The astonishingly precise manner of the routing of vesicles through the interior of a cell to various intracellular and extracellular destinations can be illustrated by numerous examples, such as the release of neurotransmitters into the presynaptic region of a nerve cell or the transport of insulin towards the cell surface. Busy cells are often filled with thousands of traveling vesicles. In the case of ribbon synapses, the number of vesicles per $\mu m^3$ with an averaged diameter of 46 nm is location-dependent and ranges between $212 \pm 69$ and $524 \pm 84$, with higher values closer to the cell membrane [6]. Membrane-bounded vesicles travel with a velocity of about 250 mm/day (i.e. 3 $\mu$m/s), while cytoskeletal proteins move only a fraction of a millimeter per day [7]. The complex mechanisms by which the vesicles form, find their correct destination, fuse with organelles or the plasma membrane and finally deliver their cargo, has been studied among others by three Laureates of the 2013 Nobel Prize in Medicine and Physiology: Schekman, Rothman and Südhof [8, 9]. Recent advances in imaging techniques allowed to examine the individual dynamics of secretory vesicles *in vitro* in more detail [10–13].

 The precise manner of the vesicle transport becomes even more impressive considering the intracellular structure of eukaryotic cells. The interior of a eukaryotic cell—the cytoplasm—is

a highly complex composite material. It is made up of the solid cytoskeleton, which is surrounded by an inhomogeneous gel-like phase, called cytosol. The cytosol itself comprises mainly water (70%), as well as many solute ions, micro- and macro-molecules such as proteins (20–30%) [14–16]. Furthermore, a variety of different not randomly distributed organelles can be found in the cytoplasm of eukaryotic cells, including the Golgi apparatus and the endoplasmic reticulum, that form a very complex structure in the vicinity of the nucleus. The endoplasmic reticulum and the Golgi apparatus are two of the main targets or starting points of transport vesicles, containing enzymes, which, e.g., fuse later on with endosomes and become lysosomes. Other important organelles are the mitochondria being responsible for the generation of most of the cell's chemical energy and are therefore mainly localized near sites of high ATP consumption.

While the organelles have numerous eminently specialized functions, the cytoskeleton on the other hand is mainly responsible for maintaining the cellular shape, strength and structural integrity. It consists of three major types of self-assembling and interconnected polymeric filaments: (i) cable-like actin filaments (diameter: 8 nm, total filament length per cell: 300 mm); (ii) pipe-like microtubules (diameter: 25 nm, total filament length per cell: 7.5-15 mm); (iii) rope-like intermediate filaments (diameter: 10 nm, total filament length per cell: 10 mm) [14]. Actin filaments, e.g., are organized completely different depending on where they are located inside the cell. In the cell cortex actin filaments are mainly organized in a 3D lattice-like network having different densities, while in the filopodium they are organized in a tight parallel bundle of 10 to 20 filaments. However, actin filaments can also be organized in so-called stress fiber bundles that may connect to the network of intermediate filaments. Recent advances in imaging techniques allow to gain detailed insights into the spatial organization, orientation and cross-linking properties of actin filaments, see, e.g., [17–21]. Microtubules usually radiate from an organizing center, called centrosome, located in the proximity of the nucleus. Intermediate filaments are reported to be often colinear with microtubules [14]. In general, the cytoskeltal network undergoes continuous growth and remodeling processes by means of filament polymerization and depolymerization in order to maintain homeostasis [22]. For a more detailed discussion on the cell biology, see [7, 23].

It is the highly heterogeneous structure of eukaryotic cells on many different scales together with the sharp contrast in biophysical properties that ensures the physiological behavior of the numerous specialized cellular functions. However, each individual intracellular component (of comparable size) serves as an obstacle that impedes the motion of vesicles within the cell. Therefore, the hypothesis is posed that any cytoskeletal alteration may directly affect the vesicle-mediated cell transport process. Recently, accessory protein candidates, signaling and control pathways have been identified in *in vitro* studies that, e.g., regulate the length or total filament content, stabilize networks by cross-linking the filaments, and are thus responsible for the reorganization of the actin cytoskeleton in response to growth factor stimulation or tumor viruses [24–27]. Zhang et al. [13] found by means of single-vesicle tracking of cell-bound membrane vesicles that the actin and microtubule polymerization inhibitors cytochalasin D and nocodazole significantly decrease the average total number, the mean speed and displacement of individual cell-bound membrane vesicles mainly due to a breakdown of the active transport mechanism. However, it remains unclear to which extend the purely diffusive part of the transport process is influenced by a disruption of the cytoskeleton. On the one hand, the active transport mechanism via motor proteins is much faster than the purely diffusive part, leading to the conclusion that functional changes of the active transport mechanism may have a more pronounced and immediate influence on the vesicle-mediated cell transport. On the other hand, a better understanding of the impact of cytoskeletal filament alterations on the purely diffusive part of the vesicle transport is particularly relevant for cells that lack an

active vesicle transport mechanism via motor proteins by nature (e.g., red blood cells) or in the case of a pathologically induced (regional) breakdown of the active transport, being the main objective of this study. Moreover, mutations or uncontrolled overactivations of actin and microtubule binding proteins are reported to be related to neurodegenerative diseases like Parkinson's and Alzheimer's disease [28], amyotrophic lateral sclerosis [29] or cancer [30]. A disruption or hyperactivity of membrane trafficking pathways is thought to contribute directly to a variety of severe neurological, immunological, liver and cardiovascular diseases, as well as intestinal, blood, renal, muscular or multi-systemic disorders, cancer and diabetes [31].

The combination of the results obtained from various biomedical investigations and detailed mechano-mathematical models with the highly efficient engineering software packages in order to simulate dynamic processes inside eukaryotic cells, such as the vesicle transport, is needed to bridge the gap between the theoretical investigations and the medical practice, and may shift the paradigm in the better understanding of different diseases. However, in contrast to the already advanced biomedical investigations, the mechano-mathematical and computational modeling of the ongoing processes within cells is rather in the initial stage, see, e.g., [32–42]. Therein, it is commonly focused on a single aspect of the biomechanics, mechanobiology or biophysics within a small intracellular domain, including certainly not enough biomedical inputs which are necessary to draw realistic conclusions. The reason therefore is simply the heterogeneity and complexity of the cytoplasm, as delineated above.

It is well known in biomechanics that morphological changes are the basis of functional changes. However, functional changes in turn may affect the morphology. Therefore, the main objective of this study was to numerically analyze the effects of specific cytoskeletal alterations onto the diffusive vesicle-mediated cell transport. In particular, the impact of the heterogeneity of the actin filament network density on the vesicle transport and the fractional anisotropy of the effective properties were investigated considering the finite size of the transport vesicles, i.e. the consideration of the inaccessible domains for the vesicles. To address the outlined objectives, this article is structured as follows: In the first section the used modeling approach is briefly summarized, containing a description of the generated parameterized 3D cell model and the implemented biophysical model for the diffusive vesicle transport which is based on the concept of homogenization owing the complexity of the cytoplasm. Results of the performed numerical investigations are presented and discussed subsequently. For several certain domains within the cytoplasm, the fractional anisotropy of the computed effective diffusion properties is investigated. Furthermore, the hypothesis is analyzed whether the relation between the effective properties and the inaccessible area fraction can be modeled by means of a two-parameter power law as suggested by Novak et al. [33]. Finally, nine different whole cell models were generated to study how alterations of the total actin filament amount triggered by a changed mechanical environment or a reduced number of filament cross-links due to a mutation of the cross-linking proteins Arp2/3 or filamin may affect the vesicle-mediated cell transport. The final section concludes with the key findings of this numerical study and provides a future outlook.

## Materials and methods

This section briefly summarizes the workflow and the methodologies used to model and analyze the passive vesicle-mediated cell transport, i.e. only the movement of the vesicles inside the cytosol hindered by the cytoskeleton. It is assumed that no external forces act on the vesicles and that only thermal fluctuations (Brownian motions) lead to a net mass transfer of vesicles. Hence, we treat the passive vesicle transport as a pure diffusion problem, since we are only interested into how the cytoskeleton affects the time dependent concentration

distribution of the vesicles inside the cell, i.e. in a closed system. A realistic parametrized three-dimensional (3D) model of the cytoskeleton was generated, which served as an input for multiscale finite element (FE) analyses of the intracellular diffusion process. The model validation was performed using Monte Carlo simulations, while the correct implementation of the proposed methodology was carefully verified by means of an example with an existing analytical solution.

## Parametrized 3D cytoskeleton model

To perform a systematic numerical study, that allows to investigate, e.g., how the actin filament network density and the associated fractional anisotropy of the network affect the vesicle transport, a parametrized 3D cytoskeleton model was generated based on recent experiments performed by Meindl et al. [43]. Therein, they compared different fluorescence-based methods (plate reader, flow cytometry and image analysis) to quantify the *in vitro* uptake of nanoparticles by phagocytes and non-phagocytic cells. Briefly, among others adenocarcinomic human alveolar basal epithelial cells (cell line A549) were cultured in Dulbecco's Modified Eagle's Medium (DMEM), 10% fetal bovine serum (FBS), 2 mM L-glutamine and 1% penicillin/streptomycin. Subsequently, the cell cultures were exposed to functionalized and non-functionalized fluorescence-labeled polysterene particles with diameters of 20 and 200 nm for 24 hours to study their individual uptake behavior. To identify the cells, and to differentiate between intracellular uptake of particles and adhesion of particles to the plasma membrane, the actin cytoskeleton was stained with Alexa Fluor® 488-phalloidin, and the nuclei were counterstained with Hoechst 33342. The cells were viewed in an A1R Laser Scan microscope (Nikon) at ex/em of 488 and 518 nm.

**Macroscopic actin density model of an A549 cell.** In the present study, image stacks of the above described experiments with a pixel resolution of $(0.2 \times 0.2 \times 0.2)$ $\mu$m were utilized and pre-processed (including noise reduction and semi-automatic segmentation) using the software package ImageJ (1.52s, National Institute of Health), such that the color intensity profiles of the three image channels (RGB) correspond to the relative density profiles of the actin filament network (green channel), the nucleus (blue channel) and the nanoparticles (red channel) for an individual A549 cell.

Afterwards, the image stacks were imported into Matlab (R2017b, The MathWorks, Inc.) and further processed. For each slice of the $z$-stack, a Gaussian mixture model $p(\mathbf{x})$, i.e. a linear superposition of $K$ Gaussians $\mathcal{N}(\mathbf{x}|\boldsymbol{\mu}_k, \boldsymbol{\Sigma}_k)$ with mean vector $\boldsymbol{\mu}_k \in \mathbb{R}^2$ and co-variance matrix $\boldsymbol{\Sigma}_k \in \mathbb{R}^{2\times2}$ weighted by a scalar mixing coefficient $0 \leq b_k \leq 1$, in the form

$$p(\mathbf{x}) = \sum_{k=1}^{K} b_k \mathcal{N}(\mathbf{x}|\boldsymbol{\mu}_k, \boldsymbol{\Sigma}_k) \qquad \text{with} \qquad \sum_{k=1}^{K} b_k = 1 \tag{1}$$

was fitted to the normalized two-dimensional (2D) color intensity profile of the actin channel $I(\mathbf{x})$ using an expectation-maximization (EM) algorithm together with a nonlinear least squares approximation [44–46]. The EM algorithm is an iterative scheme that aims to maximize the likelihood function with respect to the parameter set $(b_k, \boldsymbol{\mu}_k, \boldsymbol{\Sigma}_k)$. The algorithm is deemed to have converged when the change in the log likelihood function falls below a defined threshold.

Fig 1A shows the original color intensity distribution $I(\mathbf{x})$ of the actin channel normalized by the total density, while Fig 1B depicts the probability distribution $p(\mathbf{x})$ of the fitted Gaussian mixture model for one slice of the $z$-stack. Thereby, a minimum of $K = 557$ Gaussians was necessary to achieve a coefficient of determination of $R^2 = 0.93$. The learned Gaussian mixture model $p(\mathbf{x})$, as presented in Fig 1B, describes the experimentally obtained normalized actin

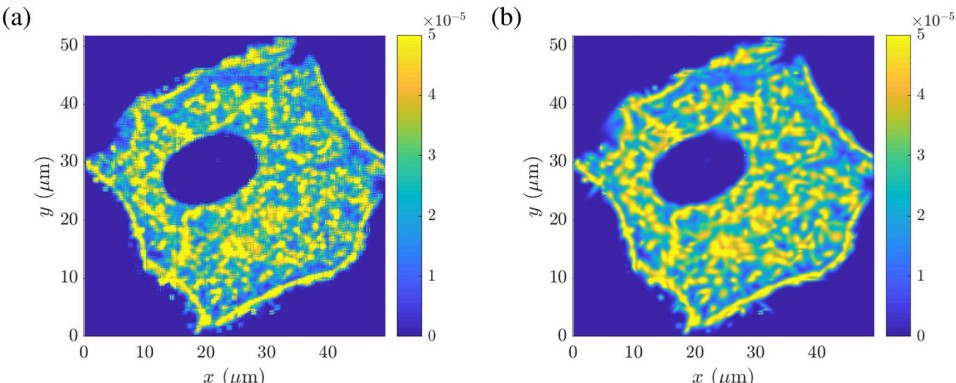

**Fig 1. Macroscopic actin density distribution of an A549 cell.** (A) Color intensity distribution $I(x, y)$ (normalized by the total intensity) of the actin channel obtained from a single slice within the image stack. (B) Corresponding probability distribution $p(x, y)$ of the fitted Gaussian mixture model using in total 557 Gaussians. Data used in this figure can be found in S1 Data.

density distribution as a continuous smooth function and is, therefore, more convenient to handle than a pixel image containing approximately 61000 discrete intensity values. Furthermore, the parameterized macroscopic model permits, e.g., to alter the density of the actin network at very specific locations by simply adapting the co-variance matrix $\Sigma_k$ or the scaling factor $b_k$ in the proximity of the region of interest.

Note that the limited resolution of the confocal microscopy images does not allow the fine details of the actin filament network to be reconstructed, such as the structure of individual filaments and their branching behavior. However, an accurate mapping of these features is the key to determine more realistic diffusion properties. Therefore, a generic 3D microscopic actin filament network was created based on the developed 2D macroscopic actin density models (obtained for each slice of the $z$-stack) in combination with published data regarding the *in vitro* spatial actin filament organization, as described in the following section.

**Microscopic actin filament network and resulting inaccessible domains.** In order to model the detailed actin network structure of an A549 cell, a Matlab routine was implemented that allows to generate a 3D actin skeleton based on the given relative macroscopic actin density distribution $p(\mathbf{x})$ for a defined position $\mathbf{x}$ within the cell, as introduced in Eq (1). Thereby, actin filaments (F-actin) were abstracted as solid cylinders having a radius of 5 nm [47–49]. The cylinders were randomly placed within a cube of size $(1 \times 1 \times 1)$ $\mu$m, oriented according to user-defined distributions. Actin filaments, grown in static conditions, are reported to be almost isotropically oriented, albeit a preference towards some specific angular orientations can be seen [50, 51]. Arp2/3-mediated actin filament branching angles of approximately 70° are reported, e.g., [17, 52, 53]. Hence, the long axes of the cylinders were uniformly distributed in $\mathbb{R}^3$ and filament branching between 65° and 75° was allowed. The absolute number of actin filaments per 1 $\mu$m$^3$ in the cell cortex is set to 1 430 with an average length of 1.1 $\mu$m according to Abraham et al. [54]. This resulted in a volume ratio of approximately 0.124, an averaged distance between neighboring filaments of ca. 40 nm and a F-actin density of 40 mg/ml, which is significantly higher than the value reported by Hartwig and Shevlin [55]. Therein, a filament concentration of 10-12 mg/ml is mentioned for the macrophage actin skeleton. Since we were only interested in the effects of relative alterations of the filament network density onto the diffusive vesicle transport, a filament density of 40 mg/ml was assigned to the maximum values of the actin density distribution $p(\mathbf{x})$, i.e. the yellow regions in Fig 1B. For all other intracellular domains the absolute number of actin filaments was scaled according to the macroscopic actin

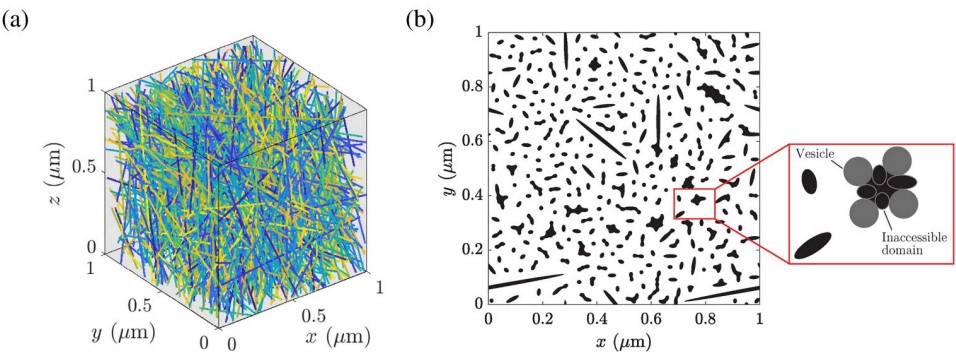

**Fig 2. Microscopic cytoskeletal network model of an A549 cell.** (A) Isotropically distributed cylindrical actin filaments in a cube of dimensions $(1 \times 1 \times 1)$ $\mu$m. (B) Horizontal cut through the 3D network at $z = 0.5$ $\mu$m with resulting inaccessible domains for vesicles having a diameter of 20 nm. The inset highlights the resulting inaccessible area (black) due to the obstacles (black with grey edges) and the finite size of the vesicles (grey).

density distribution $p(\mathbf{x})$. Fig 2A shows a generated 3D microscopic actin filament network corresponding to the position with the coordinates $x = 30$ $\mu$m and $y = 30$ $\mu$m, see Fig 1B. The network consists of 822 actin filaments with an averaged distance between neighboring filaments of 51.31 nm $\pm$ 45.29 nm (mean $\pm$ SD) and an averaged intersection angle between filaments of 71.14˚ $\pm$ 2.48˚.

To estimate the domains which are inaccessible for transport vesicles due to their finite size, sectional images were created along the $z$-direction of the generated 3D network (100 slices). Resulting image slices contained circles, ellipses and rectangles. In case the distance between neighboring obstacles is smaller than the vesicle diameter, e.g., 20 nm the passage of a vesicle via this domain is not possible and the area is treated as an additional obstacle. Therefore, too narrow obstacles were connected via undirected paths. Complex paths were simplified using graph theory, i.e. by decomposing the graphs into a tree of biconnected components, called block-cut tree, and by removing inner edges in order to reduce the maximum degree of nodes within each biconnected component to two [56]. Finally, along each path the inaccessible area within an image slice can be computed explicitly by virtually placing the vesicles (metaballs) tangentially between two neighboring obstacles. A representative result of this algorithm can be seen in Fig 2B. The corresponding inaccessible volume was then approximated by combining results of all sectional images. Thereby, the inaccessible areas between neighboring slices were averaged and multiplied by the slice distance. Hence, the generated 3D cell model allows to specifically alter the cytoskeletal actin network structure locally by simply adjusting, e.g., the filament radius, the number of filaments or the orientation distribution and cross-linking of the filaments.

Fig 3 shows the corresponding distribution of the inaccessible area fraction $\phi(\mathbf{x})$ within a horizontal 2D cut through the whole 3D cell network using the filament network parameters as defined above, such as the F-actin density, the orientation distribution, the branching angles etc. Thereby, the inaccessible area fraction was computed as the sum of the areas being inaccessible for vesicles per $\mu$m$^2$. The estimated inaccessible area fraction within the A549 cell model varied between 0 and 0.0931.

## The mathematical model problem

The passive vesicle transport in the strongly heterogeneous cytoplasm is modeled as a diffusion problem. Hence, the class of problems we are interested in are second-order linear parabolic

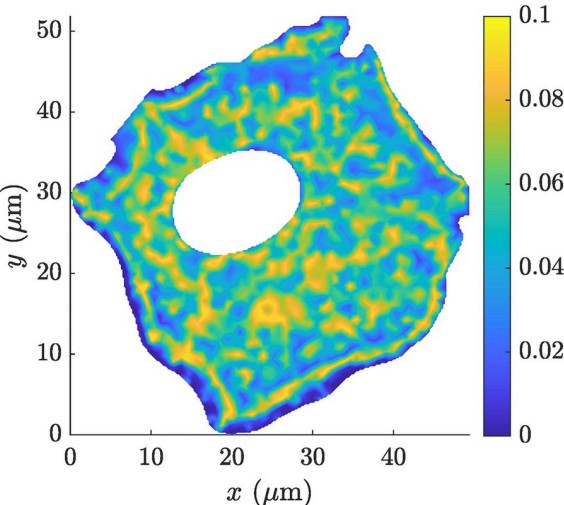

**Fig 3. Resulting inaccessible area fraction.** Distribution of the inaccessible area fraction $\phi(\mathbf{x})$ within a 2D cross-sectional cut through the whole 3D actin filament network model of the A549 cell. Data used in this figure can be found in S1 Data.

partial differential equations defined on a macroscopic (convex polygonal) domain $\Omega \in \mathbb{R}^d$, $d = \{1, 2, 3\}$, i.e. the cytoplasm, with coefficients originating from a fine microstructure. Thus, the vesicle concentration $u^\varepsilon(\mathbf{x}, t) \in [0, 1]$ defines the normalized number of vesicles per unit volume and denotes the solution of the (non-homogeneous) problem

$$
\begin{aligned}
\frac{\partial}{\partial t} u^\varepsilon(\mathbf{x}, t) - \nabla \cdot (\mathbf{D}^\varepsilon(\mathbf{x}) \nabla u^\varepsilon(\mathbf{x}, t)) &= f(\mathbf{x}), & \forall \mathbf{x} \in \Omega, t > t_0, \\
u^\varepsilon(\mathbf{x}, t) &= g_{\mathrm{D}}, & \forall \mathbf{x} \in \partial\Omega_D, t > t_0, \\
\mathbf{n} \cdot (\mathbf{D}^\varepsilon(\mathbf{x}) \nabla u^\varepsilon(\mathbf{x}, t)) &= g_{\mathrm{N}}, & \forall \mathbf{x} \in \partial\Omega_N, t > t_0, \\
u^\varepsilon(\mathbf{x}, t) &= u_0, & \forall \mathbf{x} \in \Omega, t = t_0,
\end{aligned}
\tag{2}
$$

where the vesicle concentration $g_{\mathrm{D}} \in H^{1/2}(\partial\Omega_D)$ (Dirichlet boundary condition) and the vesicle flux $g_{\mathrm{N}} \in H^{-1/2}(\partial\Omega_N)$ (Neumann boundary condition) are imposed along a Lipschitz continuous boundary $\partial\Omega = \partial\Omega_D \cup \partial\Omega_N$, i.e. the cell membrane, with $\mathbf{n}$ as the exterior normal unit vector to $\partial\Omega_N$, together with the initial vesicle concentration in the cytoplasm $u_0 \in L^2(\Omega)$. In addition, $f \in L^2(\Omega)$ is a smooth source term representing the relative number of vesicles being created or destroyed inside the cytoplasm, and $\mathbf{D}^\varepsilon = (\mathbf{D}^\varepsilon)^{\mathrm{T}} \in (L^\infty(\Omega))^{d \times d}$ is a spatially rapidly altering diffusion tensor for which $\exists \lambda, \Lambda > 0 : \lambda|\mathbf{z}|^2 \le \mathbf{z}^{\mathrm{T}} \mathbf{D}^\varepsilon \mathbf{z} \le \Lambda|\mathbf{z}|^2, \forall \mathbf{z} \in \mathbb{R}^d$, and $\forall\varepsilon$. Here, $\varepsilon$ is a small length scale parameter that signifies explicitly the multiscale nature of the diffusion tensor $\mathbf{D}^\varepsilon$ and the solution $u^\varepsilon$ corresponding to the average distance between actin filaments, as introduced in the previous section. The usual Sobolev spaces $H_D^1(\Omega) = \{v \in H^1(\Omega, v = 0 \text{ on } \partial\Omega_D\}$ with $H^1(\Omega) = \{v \in L^2(\Omega), \nabla v \in L^2(\Omega)\} = W^{1, 2}(\Omega)$ are considered. We assume that the cytoskeletal network structure does not change and that vesicles are neither created nor destroyed inside the cell during the time of interest. Thus, we treat the diffusion tensor $\mathbf{D}^\varepsilon$ and the domain $\Omega$, as well as the source term $f$ as constant with respect to time.

Numerical approximations of such a problem can be obtained by numerous techniques, e.g., the finite element method, the finite difference method, and the finite volume method. In S1 Appendix, we provide some details on the implementation and use of the standard Galerkin finite element method in space together with the backward Euler method in time.

The convergence of such a standard finite element method requires a mesh size $H$ small enough to resolve the finest length scale $\varepsilon$ of the problem, i.e. $H \ll \varepsilon$. In the case that $\varepsilon$ is small, simulations often represent prohibitive computational costs due to the necessary high number of degrees of freedom. However, neglecting the fine scale physical properties of the problem may lead to questionable results on the macroscopic scale of interest. The numerical simulation of the diffusion process in strongly heterogeneous media poses major computational challenges. Hence, so-called homogenization strategies are needed to find numerical approximations to the solution of the given model problem (Eq (2)) with a significantly reduced computational cost.

## The concept of homogenization

The theory of homogenization aims to predict the global behavior of a composite material taking into account its microscopic constituents and their properties. Thereby, a heterogeneous medium, possessing a fine not necessarily periodic microstructure with, e.g., rapidly altering coefficients or perforations, takes on the appearance of a much simpler homogeneous medium. This homogenized medium can then be modeled using a partial differential equation with effective coefficients that will capture the properties of the microstructure, supposing that the microscopic length scale is much smaller than the macroscopic size of the domain in which a certain physical phenomenon is investigated. Hence, the concept of homogenization enables to understand how the properties on the microscopic level affect the macroscopic or global behavior of a composite material.

Various different but related numerical homogenization techniques have been developed, such as the wavelet homogenization techniques by Dorobantu and Engquist [57], the multigrid numerical homogenization techniques [58, 59], the multiscale FEM (MsFEM) or the FE$^2$ method [60, 61], and the heterogeneous multiscale method (HMM) [62]. In the present study, a multiscale method is implemented in Matlab based on the HMM and the mathematical homogenization by asymptotic expansion [63]. The key aspects of the algorithm are summarized in the following subsection.

**The multiscale model for the diffusive vesice transport.** The general idea of the multiscale model is to couple solutions of numerous microscale problems in order to obtain the macroscopic behavior of the solutions. However, it is not the purpose of this method to resolve the microscale behavior in the entire domain. It can be seen as probing the local microstructure in a statistical sense.

The multiscale model for the diffusive vesicle transport contains two essential parts. The first part is an overall macroscopic scheme for the macroscopic variables on a macroscale domain $\Omega \in \mathbb{R}^d$, $d = \{1, 2, 3\}$. Hence, let $u^0(\mathbf{x})$ be the (homogenized) macroscopic vesicle concentration, i.e. the normalized number of vesicles per unit volume, denoting the solution of the (homogenized, up-scaled) macroscopic problem

$$
\begin{aligned}
\frac{\partial}{\partial t} u^0(\mathbf{x}, t) - \nabla \cdot (\mathbf{D}^0(\mathbf{x}) \nabla u^0(\mathbf{x}, t)) &= f(\mathbf{x}), &&\forall \mathbf{x} \in \Omega, t > t_0, \\
u^0(\mathbf{x}, t) &= g_D, &&\forall \mathbf{x} \in \partial\Omega_D, t > t_0, \\
\mathbf{n} \cdot (\mathbf{D}^0(\mathbf{x}) \nabla u^0(\mathbf{x}, t)) &= g_N, &&\forall \mathbf{x} \in \partial\Omega_N, t > t_0, \\
u^0(\mathbf{x}, t) &= u_0, &&\forall \mathbf{x} \in \Omega, t = t_0,
\end{aligned}
\tag{3}
$$

where the vesicle concentration $g_D \in H^{1/2}(\partial\Omega_D)$ and the vesicle flux $g_N \in H^{-1/2}(\partial\Omega_N)$ are imposed along a Lipschitz continuous boundary $\partial\Omega = \partial\Omega_D \cup \partial\Omega_N$, i.e. the cell membrane, with $\mathbf{n}$ as the exterior normal unit vector to $\partial\Omega_N$, together with the initial vesicle concentration

in the cytoplasm $u_0 \in L^2(\Omega)$. In addition, $f \in L^2(\Omega)$ is a smooth source term corresponding to the relative number of produced or destroyed vesicles in the cytoplasm, and $\mathbf{D}^0 = (\mathbf{D}^0)^{\mathrm{T}} \in (L^\infty(\Omega))^{d \times d}$ is the *a priori* unknown effective diffusion tensor representing the underlying microscopic properties at (and around) the position $\mathbf{x}$ satisfying $\exists \lambda, \Lambda > 0 : \lambda |\mathbf{z}|^2 \leq \mathbf{z}^{\mathrm{T}} \mathbf{D}^0 \mathbf{z} \leq \Lambda |\mathbf{z}|^2, \forall \mathbf{z} \in \mathbb{R}^d$. Here, the superscript 0 signifies that the corresponding variable is related to the (homogenized) macroscopic length scale. For this macroscopic problem a standard finite element method was chosen as a macroscopic solver (as introduced in S1 Appendix), where the element size of the macroscopic mesh $H$ is allowed to be larger than $\varepsilon$, see Fig 4A.

Thereby, for each macroscopic quadrature point $\hat{\mathbf{x}}_{\mathrm{ip}}$ the missing macroscopic data, i.e. the effective diffusion tensor $\mathbf{D}^0(\hat{\mathbf{x}}_{\mathrm{ip}})$, needs to be estimated. This marks the second essential part of the multiscale model of the diffusive vesicle transport and is performed by solving the original microscale model problem, sometimes called the cell problem, locally around each quadrature point $\hat{\mathbf{x}}_{\mathrm{ip}}$ within the sampling domain $K_\delta = \hat{\mathbf{x}}_{\mathrm{ip}} + \delta/2(-1, 1)^d$, $d = \{1, 2, 3\}$, and $\delta > 0$, see Fig 4B and 4C. Note that the sampling domain $K_\delta$ contains holes, which represent the inaccessible domains due to the finite vesicle size. It is assumed that vesicles are not able to penetrate the actin filaments.

From the theory of asymptotic expansion the strong formulation of the $j = \{1, \ldots, d\}$ cell problems is obtained, as presented in Allaire [63], i.e.

$$
\begin{aligned}
-\nabla \cdot (\mathbf{D}^\varepsilon(\mathbf{x}) \nabla \chi^j(\mathbf{x})) &= -\nabla \cdot (\mathbf{D}^\varepsilon(\mathbf{x}) \mathbf{e}_j), \quad \forall \mathbf{x} \in K_\delta, \\
\chi^j(\mathbf{x}), &\quad K_\delta - \text{periodic}, \\
\frac{1}{|K_\delta|} \int_{K_\delta} \chi^j \, \mathrm{d}\mathbf{x} &= 0,
\end{aligned}
\tag{4}
$$

where $\mathbf{e}_j$ are the orthonormal basis vectors of $\mathbb{R}^d$, $\mathbf{D}^\varepsilon(\mathbf{x})$ is the microscopic diffusion tensor as introduced above, and the auxiliary function $\chi^j(\mathbf{x})$ needs to be periodic in $K_\delta$. The zero mean condition of the microfunction $\chi^j(\mathbf{x})$ is necessary to obtain a unique solution. Note that the formulation of the microscopic boundary conditions is a key component in multiscale analyses. Numerical studies by Yue and Weinan [64] have shown that for a two-phase composite material, with a random checker-board distribution of the two phases, the periodic boundary

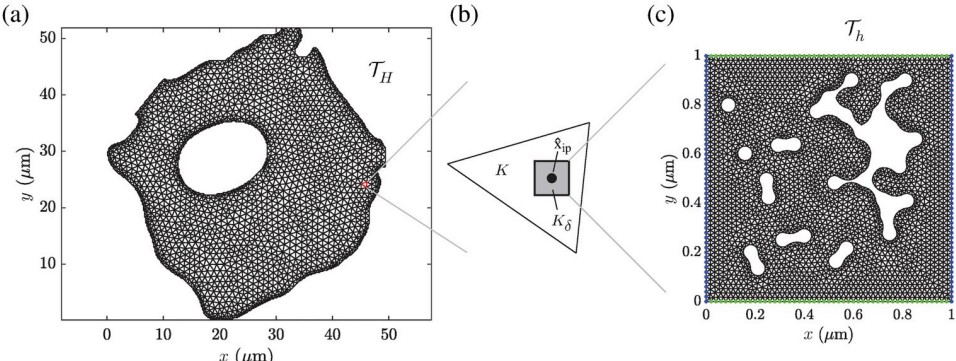

**Fig 4. Schematic of the proposed two-scale FE model for the diffusive vesicle-mediated cell transport.**
(A) Macroscopic discretization $\mathcal{T}_H$ of the domain $\Omega$ into triangular finite elements. (B) Macroscopic linear finite element $K$ with the sampling domain $K_\delta$ of size $|K_\delta| = \delta^2$ located at the quadrature point $\hat{\mathbf{x}}_{\mathrm{ip}}$. (C) Microscopic discretization $\mathcal{T}_h$ of the domain $K_\delta$ into triangular finite elements. For illustrative purposes, 30 circular obstacles with a diameter of 60 nm were positioned inside the sampling domain $K_\delta$, and the inaccessible areas were determined for vesicles with a diameter of 120 nm. The periodic boundary conditions along the edges of the sampling domain are highlighted in blue and green color.

conditions for the local microscale problems perform better than Dirichlet and Neumann boundary conditions. In general, the Neumann formulation underestimates the resulting effective tensor, while the Dirichlet formulation results in an overestimation. In S1 Appendix, the weak discretized form of the $j$ cell problems and the corresponding finite element implementation are described in detail.

Finally, the homogenized diffusion tensor $\mathbf{D}^0(\hat{\mathbf{x}}_{ip})$ at the current macroscopic quadrature point $\mathbf{x}_{ip}$ can be computed as

$$D_{ij}^0\left(\hat{\mathbf{x}}_{ip}\right) = \frac{1}{|K_\delta|} \int_{K_\delta} \left( D_{ij}^\varepsilon(\mathbf{x}) - \sum_{k=1}^d D_{ik}^\varepsilon(\mathbf{x}) \frac{\partial}{\partial x_k} \chi^j(\mathbf{x}) \right) d\mathbf{x}, \tag{5}$$

by means of the solutions $\chi^j$ of the $j$ microscopic problems, as presented in Eq (4). This computation can either be done in advance or alternatively in parallel for the quadrature points within all macroscopic finite elements, since all the microscale problems are decoupled. Hence, the homogenized diffusion problem, as introduced in Eq (3), can be solved. For problems with scale separation, this multiscale method offers substantial savings of computational costs by choosing the size of the sampling domain $K_\delta$ appropriately. Since it is assumed that the cytoskeleton stays constant over the time of interest for the vesicle transport, the homogenized diffusion tensors $\mathbf{D}^0(\hat{\mathbf{x}}_{ip})$ need to be computed only once with respect to time.

If necessary, the accuracy of the macroscopic solution $u^0(\mathbf{x})$ can be improved in a post-processing step to resolve the ongoing effects at the local microstructure. A general strategy for constructing $u^\varepsilon(\mathbf{x})$ from $u^0(\mathbf{x})$ is to use the method of local corrections, which is basically the first order asymptotic expansion, i.e.

$$u^\varepsilon(\mathbf{x}) = u^0(\mathbf{x}) - \sum_{j=1}^d \chi^j(\mathbf{x}) \frac{\partial}{\partial x_j} u^0(\mathbf{x}), \tag{6}$$

and to periodically extend it within the current element, as used in the special case of periodic homogenization.

**The energy equivalence, the macro-homogeneity or the Hill-Mandel condition.** This section aims to show that for periodic boundary conditions at the microscopic length scale the Hill-Mandel condition is fulfilled. Hence, for the thermodynamically coupled quantities, the flux $\mathbf{j} = -\mathbf{D}\nabla u$ and the concentration $u$, it must hold that

$$\begin{aligned} \mathbf{j}^0(\hat{\mathbf{x}}) \cdot \nabla u^0(\hat{\mathbf{x}}) \quad &= \frac{1}{|K_\delta|} \int_{K_\delta} \mathbf{j}^\varepsilon(\mathbf{x}) \cdot \nabla u^\varepsilon(\mathbf{x}) d\mathbf{x} \\ &= \frac{1}{|K_\delta|} \left( \int_{K_\delta} \mathbf{j}^\varepsilon(\mathbf{x}) d\mathbf{x} - \int_{K_\delta} \mathbf{j}^\varepsilon(\mathbf{x}) \mathbf{J}^T d\mathbf{x} \right) \cdot \nabla u^0(\hat{\mathbf{x}}), \end{aligned} \tag{7}$$

at each macroscopic point $\hat{\mathbf{x}} \in \Omega$ with the linearized derivative of Eq (6), i.e.

$$\nabla u^\varepsilon(\mathbf{x}) = \nabla u^0 - \sum_{j=1}^d \nabla \chi^j(\mathbf{x}) \frac{\partial}{\partial x_j} u^0 = (\mathbf{I} - \mathbf{J}^T)\nabla u^0, \tag{8}$$

where $J_{ij} = \partial \chi^i(\mathbf{x})/\partial x_j$. The first integral in Eq (7) yields the definition of the macroscopic flux $\mathbf{j}^0$ as the volume average of the microscopic flux $\mathbf{j}^\varepsilon$, which is given as

$$\mathbf{j}^0(\hat{\mathbf{x}}) = -\mathbf{D}^0(\hat{\mathbf{x}})\nabla u^0(\hat{\mathbf{x}}) = -\frac{1}{|K_\delta|} \int_{K_\delta} \mathbf{D}^\varepsilon(\mathbf{x})\nabla u^\varepsilon(\mathbf{x}) d\mathbf{x} = \frac{1}{|K_\delta|} \int_{K_\delta} \mathbf{j}^\varepsilon(\mathbf{x}) d\mathbf{x}, \tag{9}$$

using Eqs (5) and (8), whereas the second integral in Eq (7) needs to be zero in order that

the Hill-Mandel condition holds true. By means of Green's first identity and by introducing $q(\mathbf{x}) = \mathbf{j}^{\varepsilon}(\mathbf{x}) \cdot \mathbf{n}$ as the normal component of the flux $\mathbf{j}^{\varepsilon}(\mathbf{x})$, where $\mathbf{n}$ is the exterior normal unit vector to $\partial K_{\delta}$, we obtain

$$
\begin{aligned}
\int_{K_{\delta}} \mathbf{j}^{\varepsilon}(\mathbf{x}) \mathbf{J}^{\mathrm{T}} \cdot \nabla u^0 \mathrm{d}\mathbf{x} \quad = \quad & \left( - \int_{K_{\delta}} [\chi^1(\mathbf{x}), \chi^2(\mathbf{x})]^{\mathrm{T}} \nabla \cdot \mathbf{j}^{\varepsilon}(\mathbf{x}) \mathrm{d}\mathbf{x} \right. \\
& \left. + \int_{\partial K_{\delta}} [\chi^1(\mathbf{x}), \chi^2(\mathbf{x})]^{\mathrm{T}} q(\mathbf{x}) \, \mathrm{d}s_{\mathbf{x}} \right) \cdot \nabla u^0 = 0,
\end{aligned}
\tag{10}
$$

for a two-dimensional problem, which holds if the continuity equation on the sampling domain $\nabla \cdot \mathbf{j}^{\varepsilon}(\mathbf{x}) = 0$ is satisfied, the microfluctuations $\chi^i(\mathbf{x})$ are periodic, i.e. $\chi^i(\mathbf{x})|_{\partial K_{\delta}^+} = \chi^i(\mathbf{x})|_{\partial K_{\delta}^-}$ on opposite boundaries $\partial K_{\delta}^+$ and $\partial K_{\delta}^-$ of the sampling domain $K_{\delta}$, as well as the normal components of the flux $q(\mathbf{x})$ are anti-periodic on opposite sites of the sampling domain $\partial K_{\delta}$, i.e. $q(\mathbf{x})|_{\partial K_{\delta}^+} = -q(\mathbf{x})|_{\partial K_{\delta}^-}$. Hence, it can be seen that for periodic boundary conditions at the microscopic scale the Hill-Mandel condition, as given in Eq (7), is fulfilled.

## Validation and verification

The effective diffusion coefficients $D_{ij}^0$, computed by the heterogeneous multiscale method at the sampling domain $K_{\delta}$, were compared with the coefficients $\tilde{D}_{ij}^0$ obtained by a Monte Carlo simulation of vesicles undergoing random walks on the same domain $K_{\delta}$, according to Novak et al. [33, 65], Bressloff and Newby [42]. Furthermore, the implementation of the proposed multiscale method was verified by means of a one-dimensional steady-state diffusion problem with an oscillatory diffusion coefficient originating from a fine scale $\varepsilon$, for which an analytical solution can be computed. A detailed description and results of the validation and verification process are presented in S2 Appendix.

## Results

Based on the developed parametrized 3D actin cytoskeleton model, the proposed method was used to analyze the computed effective diffusion tensor as a function of the inaccessible area fraction, and to quantify how alterations of the actin network density may affect the diffusive vesicle-mediated transport within an A549 cell. All simulations were performed under isothermal conditions at 37˚C. Furthermore, it was assumed that the temperature is homogeneously distributed throughout the cell.

### Approximating the effective diffusion coefficients by a simple power law

Novak et al. [33] have shown that the dependence between the effective diffusion tensor obtained by numerical homogenization and the inaccessible volume fraction is accurately described by means of a two-parameter power law for a variety of differently shaped overlapping obstacles (spheres, cylinders and disks) randomly placed in a 3D sampling domain. By assuming that an almost isotropic effective diffusion tensor $\mathbf{D}^0$ is obtained, Novak et al. [33] proposed the relationship

$$
\frac{\mathrm{tr}\mathbf{D}^0/d}{D_{\text{in vitro}}} = \frac{(1 - \phi/\phi_{\mathrm{c}})^{\mu}}{1 - \phi},
\tag{11}
$$

where $d$ is the dimension of the problem, $D_{\text{in vitro}}$ is the diffusion coefficient of the isotropic solvent, $\phi$ is the inaccessible area fraction due to the obstacles and the finite size of the vesicles, $\phi_{\mathrm{c}}$ is the critical inaccessible area fraction at which the vesicles are trapped by the obstacles,

and $\mu$ is an empirical constant. To study whether Eq (11) is a good approximation for the diffusion of vesicles having a radius of $R = 10$ nm within the cytoplasm, a set of microscopic actin filament network models was generated having different filament densities. For each network model, the inaccessible domains within a single 2D slice of the 3D network model were estimated, as described above and shown in Fig 2. Afterwards, the resulting 2D sampling domains were discretized using linear triangular finite elements and the effective diffusion tensor $\mathbf{D}^0$ was computed for each sampling domain, according to Fig 4. The mesh size of the microscopic triangulation was chosen such that the obtained effective diffusion coefficients exhibited a converged behavior. The *in vitro* diffusion tensor of the cytosol was assumed to be isotropic, i.e. $\mathbf{D}_0 = D_{\text{in vitro}}\mathbf{I}$. For spherical vesicles with a radius $R = 10$ nm, the *in vitro* diffusion coefficient estimated by the Stokes–Einstein relation is given as $D_{\text{in vitro}} = k_{\text{B}}T/(6\pi\eta R) = 32.8\mu\text{m}^2/\text{s}$ assuming that the viscosity of the cytoplasm is roughly the same as that of pure water at a temperature of $37°$C, i.e. $\eta \approx \eta_{\text{water}} = 0.692$ mPas at $T = 310.15$ K, and the Boltzmann's constant $k_{\text{B}} = 1.34 \cdot 10^{-23}$ J/K.

The resulting diffusion coefficients of the tensor $\mathbf{D}^0$ were collected and analyzed with respect to the inaccessible area fraction $\phi$. In Fig 5 it can be seen that the diffusion coefficients $D_{11}^0$ and $D_{22}^0$ were both monotonically decreasing with increasing inaccessible area fractions. The values of $D_{12}^0$ were close to 0 for all computed effective diffusion tensors. Furthermore, Fig 5 highlights that the obtained relative effective diffusion coefficients $D_{11}^0$ and $D_{22}^0$ can be approximated accurately by means of Eq (11) with $\phi_{\text{c}} = 0.6209$ and $\mu = 1.183$ corresponding to the experimentally observed obstacle geometries in an A549 cell, which is in relatively good agreement with the values presented in Novak et al. [65] for the effective diffusion in 2D amid elliptical obstacles with different aspect ratios. The parameters $\phi_{\text{c}}$ and $\mu$ were estimated by means of a nonlinear least squares approximation with the restrictions $0 \le \phi_{\text{c}} \le 1$ and $\mu \ge 0$. An $R^2$ value of 0.9999 was achieved. Considering the obtained values of $\phi$, as presented in Fig 3, the effective diffusion coefficients are reduced only by maximum 10% due to the *in vitro* actin filament network compared to the unobstructed diffusion case.

In addition, the anisotropy of the obtained diffusion tensors $\mathbf{D}^0$ was analyzed by means of the fractional anisotropy, defined as $\text{FA} = (\lambda_1 - \lambda_2)/\sqrt{\lambda_1^2 + \lambda_2^2}$, where $\lambda_1$ and $\lambda_2$ are the two eigenvalues of $\mathbf{D}^0$ with $\lambda_1 \ge \lambda_2$. Note that a fractional anisotropy close to zero means isotropic

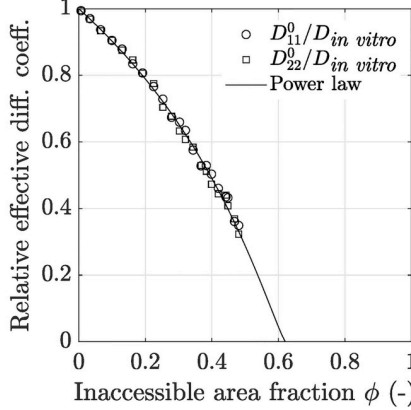

**Fig 5. Relationship between the effective diffusion coefficients and the inaccessible area fraction.** Some representative relative effective diffusion coefficients (normalized by the *in vitro* diffusion coefficient $D_{\text{in vitro}}$) for experimentally observed obstacle geometries in eukarytic cells obtained by the homogenization strategy as well as the approximation by means of a power law (see Eq (11)). Data used in this figure can be found in S1 Data.

**Table 1. Selected effective diffusion coefficients $D_{11}^0$, $D_{22}^0$ and $D_{12}^0$ normalized by the diffusion coefficient of the cytosol $D_{\text{in vitro}}$ for different realistic sampling domains having inaccessible area fractions $\phi$.** The anisotropy of the obtained diffusion tensors is analyzed by means of the fractional anisotropy (FA). Furthermore, the number of used elements and degrees of freedom (DOFs), as well as the required computation time are presented.

| $\phi$ (-) | Elements | DOFs | $D_{11}^0/D_{\text{in vitro}}$ (-) | $D_{22}^0/D_{\text{in vitro}}$ (-) | $D_{12}^0/D_{\text{in vitro}}$ (-) | FA (-) | Comp. time (min) |
|---|---|---|---|---|---|---|---|
| 0.0336 | 21 620 | 11 263 | 0.9703 | 0.9689 | 0.0015 | 0.0024 | 17.5 |
| 0.0659 | 21 261 | 11 313 | 0.9368 | 0.9336 | −0.0011 | 0.0030 | 17.6 |
| 0.0996 | 21 122 | 11 494 | 0.9049 | 0.9049 | −0.0019 | 0.0030 | 18.4 |
| 0.1622 | 20 597 | 11 659 | 0.8347 | 0.8461 | −0.0075 | 0.0159 | 19.2 |
| 0.2252 | 19 892 | 11 731 | 0.7664 | 0.7752 | 0.0010 | 0.0083 | 19.6 |
| 0.3029 | 18 821 | 11 646 | 0.6594 | 0.6327 | 0.0048 | 0.0311 | 18.9 |
| 0.3991 | 17 473 | 11 501 | 0.5032 | 0.4728 | 0.0048 | 0.0553 | 18.5 |
| 0.4808 | 16 172 | 11 177 | 0.3490 | 0.3229 | 0.0210 | 0.1038 | 16.9 |

diffusion, whereas anisotropy is given if one of the eigenvalues is higher than the other. Results in Table 1 show that the obtained diffusion tensors are in general almost perfectly isotropic. However, with increasing inaccessible area fractions the fractional anisotropy is marginally increasing, compare also with Fig 5. Hence, the assumption $D_{11}^0 = D_{22}^0$ and $D_{12}^0 = 0$ seems to be legit, especially in the domain $\phi < 0.4$. Therefore, the effective diffusion tensor $\mathbf{D}^0$ can be fully determined by Eq (11).

Fig 6 indicates a strong impact of the vesicle radius onto the inaccessible area fraction for a sampling domain $K_\delta$ having a specific actin filament density. It can be seen that the diffusion of vesicles through an actin filament network is sensitive to the vesicle size, especially if the vesicle radius is comparable to or greater than the filament radius ($R \geq 5$ nm). With further increasing vesicle radius $R$, the inaccessible area fraction approaches the maximum $\phi = 1$, whereby the slope of the sigmoidal function depends on the filament network density. Less dense networks exhibit a less steep increase of the inaccessible area fraction with increasing vesicle size. By means of Eq (11), the corresponding relative effective diffusion coefficient can be estimated. Note that for inaccessible area fractions $\phi \geq \phi_c$ the vesicles are trapped by the

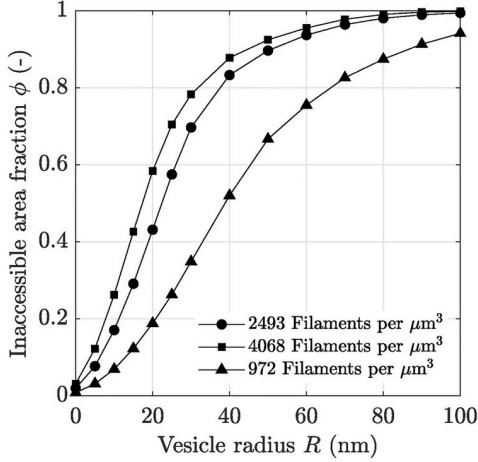

**Fig 6. Impact of the vesicle size on the inaccessible area fraction.** Resulting inaccessible area fraction $\phi$ as a function of the vesicle radius $R$ for three different 2D sampling domains cut out of the 3D cytoskeletal actin network model containing a specific amount of filaments per $\mu m^3$, compare also with Fig 2. Data used in this figure can be found in S1 Data.

filament network, whereby $\phi_c = 0.6209$. Hence, the maximum vesicle radius for the passive vesicle transport can be estimated. Assuming a maximum number of 1430 actin filaments per $\mu m^3$ in the cell cortex, a maximum vesicle diameter of approximately 90 nm is predicted, which corresponds well to experimentally observed values published by Vigers et al. [2].

The computation of the effective properties of a single sampling domain took approximately 18.3 minutes on average. One key aspect of the present study was to compute the required traveling time of vesicles to reach a specific location within an A549 cell, e.g., the Golgi apparatus or the endoplasmic reticulum located in the proximity of the nucleus, and how this traveling time is affected by filament density alterations. Therefore, the macroscopic A549 cell model has to be discretized by means of finite elements and an effective diffusion tensor needs to be computed for each macroscopic element due to the highly heterogeneous distribution of the actin filament density, see Fig 1. Discretizing the macroscopic cell model into approximately 4000 finite elements, see Fig 4A, and using the proposed homogenization strategy is still prohibitively expensive from a computational point of view even if it is run in parallel mode. Hence, Eq (11) is used for the subsequent simulations to predict the effective diffusion tensors $\mathbf{D}^0(\mathbf{x})$ on the basis of the inaccessible area fraction $\phi(\mathbf{x})$, as shown in Fig 3.

## The influence of cytoskeletal density alterations on the diffusive vesicle transport in A549 cells

The cytoskeletal filament network is known to be able to adapt in response to changed loading conditions or genetic defects and disorders. This includes alterations of the number of filaments per $\mu m^3$ and the cross-linking capability, which both influence the actin filament density of the cytoskeletal network. A sustained increase of equi-biaxial mechanical loading may lead to a homogeneously distributed elevation of the filament density. On the other hand, a genetic disorder of the cross-linking protein Arp2/3 leads to sparser filament networks. To analyze the effects of actin filament density alterations on the diffusive vesicle-mediated transport mechanisms inside an A549 cell, the (normal) *in vitro* actin filament density distribution $p(\mathbf{x})$, as introduced in Eq (1), with the fitted parameter set $(b_k, \mathbf{\Sigma}_k, \boldsymbol{\mu}_k)$ according to Fig 1B, was altered by systematically changing the parameters $b_k$ and $\mathbf{\Sigma}_k$. Hence, the modified density distribution reads $p(\mathbf{x}) = \sum_k \bar{b}_k \mathcal{N}(\mathbf{x}|\boldsymbol{\mu}_k, \bar{\mathbf{\Sigma}}_k)$ with $k = \{1, \ldots, K\}$. While the parameter $b_k$ is a purely multiplicative scaling factor, an increase of the coefficients in $\mathbf{\Sigma}_k$ leads to a higher dispersion but also to a decrease in the peak values of the function $p(\mathbf{x})$ and *vice versa*. Thus, nine

**Table 2. Impact of alterations of the parameters $b_k$ and $\Sigma_k$ in Eq (1) onto the maximum number of filaments per $\mu m^3$, the cumulative filament density (i.e. summation over $\Omega$), the cumulative inaccessible area fraction, and the time which the vesicles need to generate a concentration $u^0(\mathbf{x_A}) = 0.9$ at the position $\mathbf{x_A} = [30, 30]^T$ $\mu$m within the A549 cell.** All values are presented as relative differences measured and normalized with respect to the *in vitro* case, i.e. $\bar{\mathbf{\Sigma}}_k = \mathbf{\Sigma}_k$ and $\bar{b}_k = b_k$.

| Case | Parameters | | Relative changes with respect to the *in vitro* case (%) | | | |
|------|-----------|-----------|-------------------------|----------------------|-----------------------------|------------------------------|
| | $b_k$ | $\bar{\mathbf{\Sigma}}_k$ | Max. filaments per $\mu m^3$ | Cum. filament density | Cum. inaccessiblearea fraction | Required time for $u^0(\mathbf{x_A}) = 0.9$ |
| I | $0.5b_k$ | $\mathbf{\Sigma}_k$ | −50.0 | −50.0 | −49.2 | −2.90 |
| II | $2b_k$ | | + 100.0 | + 100.0 | + 93.8 | + 6.26 |
| III | $5b_k$ | | + 400.0 | + 400.0 | + 337.7 | + 35.00 |
| IV | $b_k$ | $0.5\mathbf{\Sigma}_k$ | + 78.3 | −2.5 | −3.5 | + 0.16 |
| V | | $2\mathbf{\Sigma}_k$ | −1.9 | + 0.4 | + 0.8 | −0.09 |
| VI | | $5\mathbf{\Sigma}_k$ | −13.6 | −1.2 | −0.7 | −0.13 |
| VII | $5b_k$ | $0.5\mathbf{\Sigma}_k$ | + 791.1 | + 387.3 | + 301.1 | + 39.3 |
| VIII | | $2\mathbf{\Sigma}_k$ | + 390.3 | + 402.2 | + 347.9 | + 33.1 |
| IX | | $5\mathbf{\Sigma}_k$ | + 332.2 | + 393.8 | + 346.4 | + 30.8 |

different density distribution functions $p(\mathbf{x})$ were generated in total and analyzed with respect to the *in vitro* case, see Table 2.

Therefore, the macroscopic A549 cell geometry was discretized into 4074 linear triangular finite elements with a maximum mesh element size of 1 $\mu$m, i.e. the length of the longest edge in the mesh, see Fig 4A. For each macroscopic integration point, a microscopic actin filament network was generated on the basis of the given filament density distribution $p(\mathbf{x})$, and the resulting inaccessible area fraction $\phi(\mathbf{x})$ was computed for a single 2D cut (sampling domain) through the 3D microscopic network, as described above. Again, the vesicle radius was set to $R = 10$ nm thereby. The resulting distributions of the inaccessible area fraction $\phi(\mathbf{x})$ are

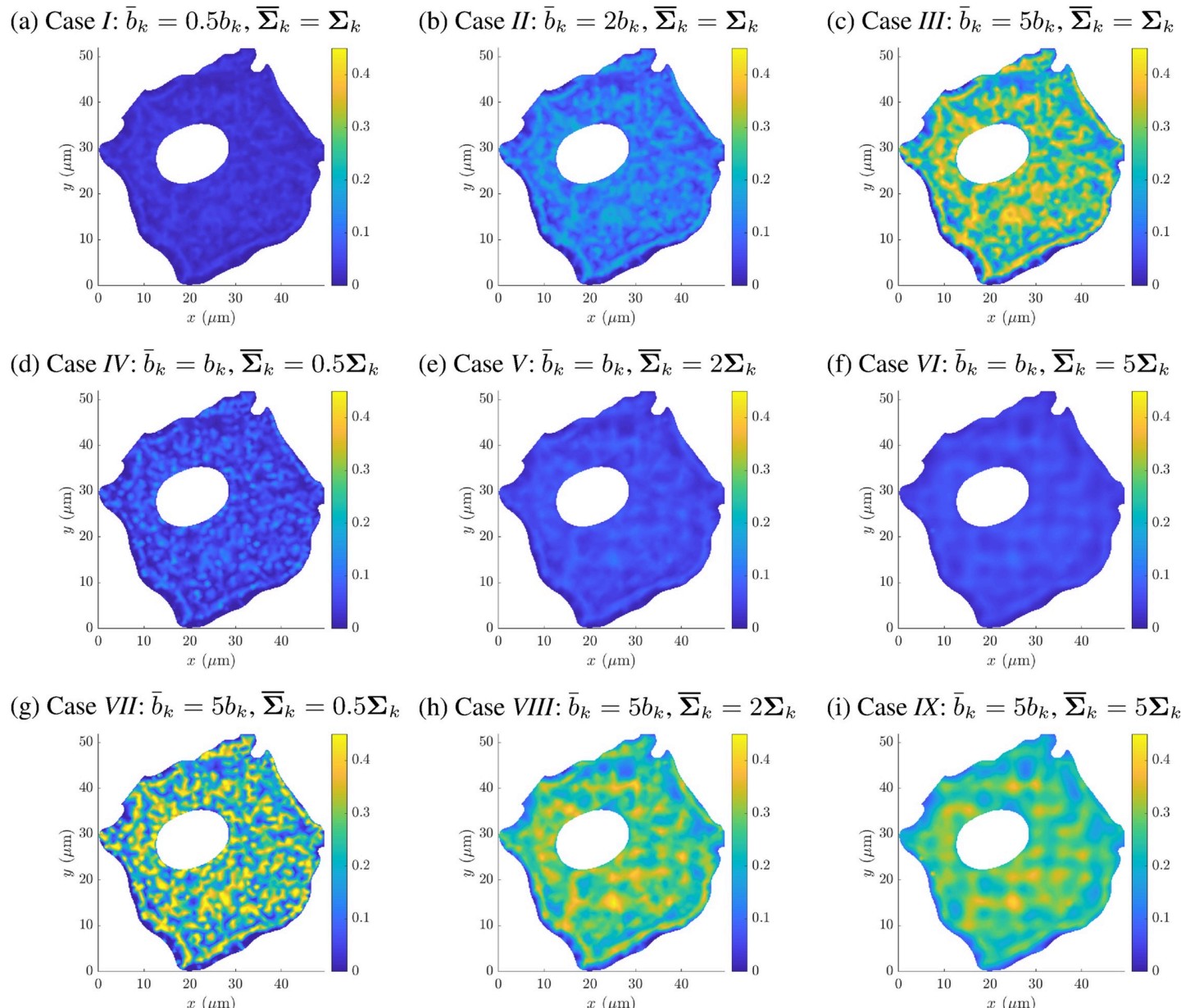

**Fig 7. Resulting inaccessible area distributions for different actin filament distributions.** Influence of modifications of the fitted parameters $b_k$ and $\mathbf{\Sigma}_k$ in Eq (1) onto the distribution of the area fraction $\phi(\mathbf{x})$ for nine cases, (A)–(I), see Table 2. Compare with the *in vitro* distribution in Fig 3. Data used in this figure can be found in S1 Data.

shown in Fig 7. As expected, an increasing $b_k$ leads to an almost linear increase of the inaccessible area fractions due to the uniformly rising filament density, see Fig 7A–7C. Note that with increasing coefficients in $\mathbf{\Sigma}_k$ the domains with relatively high inaccessible area fractions are damped while the area fractions in the surroundings are slightly lifted, especially in the proximity of the cell membrane, compare with Fig 7D–7F. This smoothing effect gets even more emphasized when the parameter $b_k$ is increased four-fold and the coefficients in $\mathbf{\Sigma}_k$ are gradually increased, as shown in Fig 7G–7I. The relative changes of the maximum number of filaments per $\mu m^3$, the cumulative filament density, and the cumulative inaccessible area fraction computed with respect to the *in vitro* case are summarized in Table 2.

Thus, to study the impact of the different actin filament density distribution models $p(\mathbf{x})$ onto the required traveling time which the vesicles need to generate a concentration $u^0(\mathbf{x}_A) = 0.9$ at the position $[\mathbf{x}_A] = [30, 30]^T \, \mu m$, the passive vesicle-mediated transport in the strongly heterogeneous cytoplasm of the A549 cell is modeled as a time dependent diffusion problem, as introduced in Eq (3). It is assumed that the vesicles are constantly taken up by the cell during the time of interest such that the vesicle concentration along the whole cell membrane is defined as $g_D = 1$. Furthermore, it is assumed that there were initially no vesicles inside the cell and vesicles are neither created nor destroyed inside the cell during the time of interest, i.e. $u_0 = 0$ and $f(\mathbf{x}) = 0, \forall \mathbf{x} \in \Omega$. For each case, the distribution of the effective diffusion tensor $\mathbf{D}^0(\mathbf{x}) = D^0(\mathbf{x}) \mathbf{I}$ was estimated by means of Eq (11), assuming that the *in vitro* diffusion tensor of the cytosol is isotropic, i.e. $\mathbf{D}_0 = D_{\text{in vitro}} \mathbf{I}$, with $D_{\text{in vitro}} = 32.8 \, \mu m^2/s$, as used above.

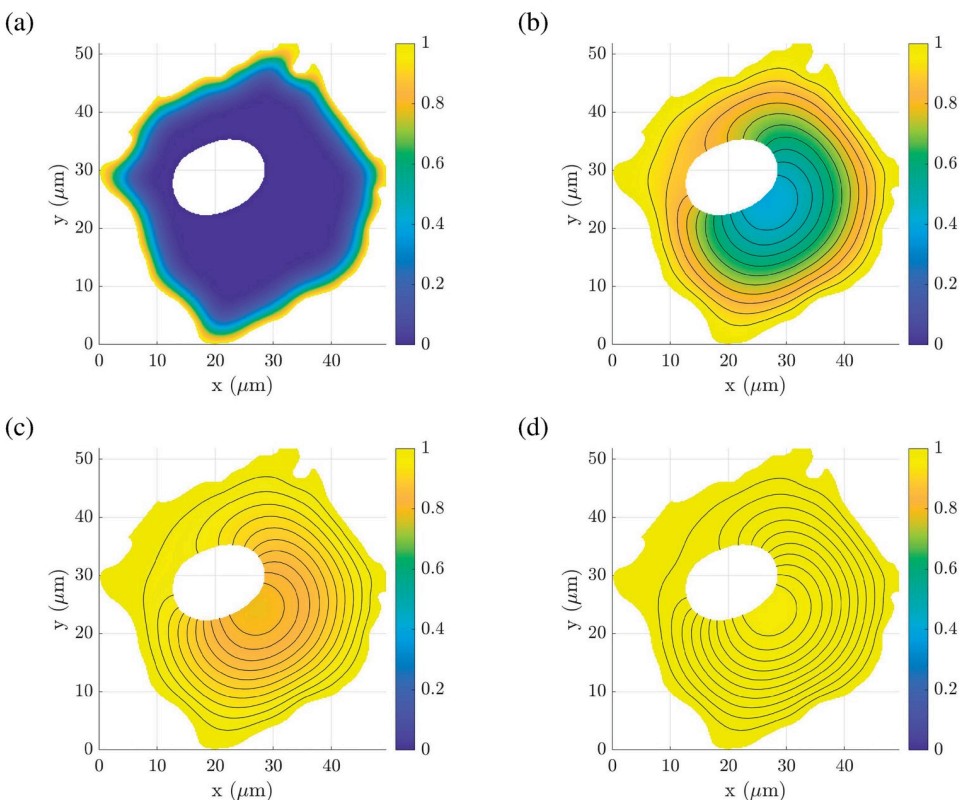

**Fig 8. Spatial distribution of the vesicle concentration $u^0(x, y)$ for the *in vitro* actin filament density distribution plotted at different points in time.** (A) At $t = 0.1$ seconds. (B) At $t = 2.5$ seconds. (C) At $t = 5$ seconds. (D) At $t = 10$ seconds. It is assumed that there were initially no vesicles inside the cell and that vesicle are constantly taken up by the cell during the time of interest such that the vesicle concentration along the cell membrane equals to one throughout the simulation. Data used in this figure can be found in S1 Data.

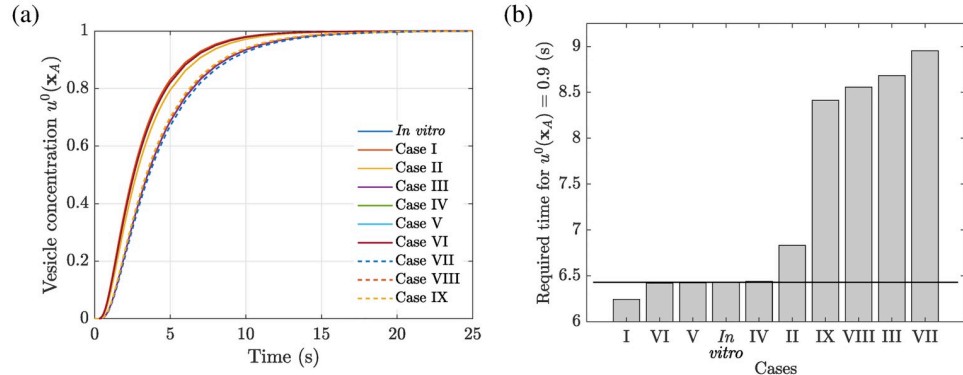

**Fig 9. Impact of cytoskeletal actin filament density alterations onto the diffusive vesicle-mediated transport in an A549 cell studied by means of ten different cases in total, see Table 2.** (A) Time evolution of the vesicle concentration $u^0(\mathbf{x}_A)$ at position $[\mathbf{x}_A] = [30, 30]^T$ μm. (B) Required traveling time which the vesicles need to generate a concentration of $u^0 = 0.9$ at position $\mathbf{x}_A$. The horizontal line denotes the traveling time of the vesicles for the *in vitro* actin filament density distribution. Data used in this figure can be found in S1 Data.

Fig 8 depicts the smooth spatial distribution of the homogenized vesicle concentration $u^0(\mathbf{x})$ for different points in time between 0.1 and 10 seconds of the diffusion process in the A549 with an *in vitro* actin filament density distribution. The time evolution of the vesicle concentration at position $\mathbf{x}_A$, i.e. in the proximity of the nucleus, is presented in Fig 9A for the *in vitro* case along with the nine different cases, *I* to *IX*, as introduced in Table 2. Note that cases *III* and *VII* to *IX* exhibit a significantly flatter slope compared to the other cases. All cases reach a steady-state value after approximately 25 seconds. The numerical simulations have shown that for the *in vitro* case it takes approximately 6.4 seconds until a vesicle concentration of 90% is reached at the position $\mathbf{x}_A$, as can be seen in Fig 9B in comparison to required traveling times for the cases *I* to *IX*. The corresponding relative time differences for the nine studied cases are presented in Table 2.

It can be seen that the required time increases monotonically with increasing $b_k$ and constant $\boldsymbol{\Sigma}_k$, i.e. the cases *I* to *III*. Since $b_k$ is a purely multiplicative scaling factor, a four-fold increase of $b_k$ leads to a four-fold increase of the maximum number of filaments per μm³ as well as the cumulative filament density. This uniformly increased filament density may, e.g., be related to an altered mechanical environment. The cell may have sensed significantly higher stresses and stretches, and therefore mechanotransduction has led to an increase of the actin filament density and the cross-linking capacity in order to reach a homeostatic stress level again. Hence, the cumulative inaccessible area fraction is increased by 337.7% for case *III* which results finally in a 35% increase of the required traveling time compared to the *in vitro* case. Case *II* follows the same line of reasoning, while case *I* exhibits a reduced traveling time due to a depolymerization of the actin network, e.g., as a response to a reduced mechanical loading state.

However, the same relative alterations of the coefficients in $\boldsymbol{\Sigma}_k$ revealed only insignificant changes of the traveling time when $b_k$ is kept constant, i.e. the cases *IV* to *VI*. The reason for this is that the parameter $\boldsymbol{\Sigma}_k$ modifies not only the width of the corresponding density distribution but also its height. Hence, a 50% decrease of the coefficients in $\boldsymbol{\Sigma}_k$ (see case *IV*) exhibits a 78.3% increase of the maximum number of filaments per μm³. On the other hand, the cumulative filament density was slightly decreased by 2.5%, and the cumulative inaccessible area fraction by 3.5%. For increasing coefficients in $\boldsymbol{\Sigma}_k$ and constant $b_k$, the maximum number of filaments assigned per μm³ decreases. Note that even though the cumulative filament density

and the cumulative inaccessible area fraction are slightly increased for a one-fold increase of $\Sigma_k$ (case *V*), the required traveling time is reduced by 0.09%. A further increase of $\Sigma_k$ (case *VI*) revealed a decrease of the cumulative filament density, the area fraction and the required traveling time compared to the *in vitro* case. In general, for uniformly increasing coefficients in $\Sigma_k$ and for keeping $b_k$ constant, the traveling time tends to slightly decrease.

This phenomenon gets more pronounced when $b_k$ is increased four-fold and $\Sigma_k$ is altered by the same amounts as before, see cases *VII* to *IX*. While the total number of actin filaments within the whole A549 cell model remains almost constant (an approximately four-fold increase compared to the *in vitro* case), the required traveling time decreases with increasing $\Sigma_k$ compared to the case *VII*. This can be explained by the fact that macroscopic diffusion barriers are smoothed out by keeping the total number of filaments almost constant, compare also with the distributions of the inaccessible area fraction in Fig 7G–7I. Hence, locally very dense filament networks become sparser, which means that the same amount of filaments is distributed over a larger domain compared to the case *VII*. This can, e.g., be related to a reduced capability to form cross-links due to genetic defects of the cross-linking proteins Arp2/3 or filamin. However, an overactivation of these proteins would lead to significantly denser networks and increased traveling times.

## Discussion

The vesicle-mediated cell transport plays an important role in the metabolism, the transport and the temporary storage of food and enzymes. Vesicles also serve as chemical reaction chambers inside the eukaryotic cells. Hence, sustained alterations of the precise manner of the vesicle transport mechanism may directly lead to pathological consequences on the cellular, tissue and organ level [31]. In general, the motion of vesicles inside the cytoplasm is a rather complex phenomenon. They either bind to motor proteins and move along microtubules or undergo hindered diffusion in the gel-like heterogeneous cytosol which is surrounded by a complex 3D cytoskeletal network, where they experience and exert thermal and hydrodynamic forces. Because the cytoskeleton provides the cell with strength and resilience, the cytoskeleton is constantly reorganized, triggered by growth factors in order to maintain homeostasis [25, 27]. This growth and remodeling processes may include modifications of the total filament amount and the cross-linking capability leading to alterations of the filament network density and porosity or the filament orientation. However, changes of the mechanical environment most likely also trigger adaptations of the surrounding extracellular matrix in soft tissues. Thus, it is hypothesized that increases of the cytoskeletal filament density slow down the diffusive vesicle-mediated cell transport and, therefore, affect important physiological cellular and tissue maintenance processes such as the synthesis of collagen, whereby the collagen pre-cursors are transported by means of vesicles from the endoplasmic reticulum to the Golgi apparatus and finally undergo exocytosis [7, 23]. Furthermore, a significant increase of the cortical actin network may slow down the release of neurotransmitters into the presynaptic region of nerve cells, resulting in a delayed signal transduction. These examples already highlight the clinical relevance and importance of investigating the impact of cytoskeletal filament alterations onto the vesicle-mediated cell transport in order to gain better insights into possible causes and consequences of many related diseases and disorders, such as Parkinsons's and Alzheimer's disease, amyotrophic lateral sclerosis, and a variety of severe neurological, immunological, liver and cardiovascular diseases, as well as intestinal, blood, renal, muscular or multi-systemic disorders, cancer and diabetes. Given that the motor protein-assisted vesicle transport occurs typically over large distances and is much faster than the diffusive vesicle transport, functional changes of the motor proteins or any structural adaptations of the microtubule network may

have a more pronounced impact on the intracellular transport processes. Nevertheless, the influence of cytoskeletal alterations on the purely diffusive part of the vesicle-mediated cell transport must not be underestimated, in particular, related to cells with no active transport mechanism by nature or cells with functional disorders of the active transport components.

In the present study we addressed this issue by means of a systematic numerical approach focusing on the diffusive part of the vesicle transport within a eukaryotic cell. A realistic parametrized 3D geometrical model of the actin filament network within an A549 cell was generated, based on an experimental study of Meindl et al. [43] and additional microstructural information about the actin filament organization. The implemented code allowed to alter the filament density and orientation distribution, the filament branching angles, and finally, it considered the excluded volume effect due to the finite size of the vesicles. Due to a lack of microstructural information, other cytoskeletal filaments and organelles were neglected. Thus, in the present study we focused purely on the actin filament network. Since actin filaments occupy almost twice as much volume as intermediate filaments and microtubules in a typical eukaryotic cell, it is assumed that alterations of the actin filament organization have a more considerable impact on the diffusive properties [14].

While a detailed consideration of all the structural complexities within the cytoplasm would yield prohibitive computational costs when using standard numerical methods, a naive disregard of the fine scale features would lead to questionable results even on the macroscopic (cellular) scale. To overcome this problem, a multiscale model for the diffusive vesicle transport was implemented on the basis of the concept of homogenization. Numerical studies revealed that the effective diffusion coefficients obtained by the proposed multiscale method are in good agreement with the results obtained by a purely stochastic method, the Monte Carlo method [42], which is in line with the findings of Novak et al. [33] and Donovan et al. [66]. Due to the significantly smaller computational costs, the proposed multiscale method seems to be a valuable tool to study the impact of various cytoskeletal alterations onto the diffusivity within the cytoplasm of a whole eukaryotic cell. However, it has been shown that in multiscale FE analyzes, the size of the sampling domain as well as the micro- and macroscopic mesh sizes need to be chosen carefully to obtain good approximations of the effective properties and an accurate homogenized solution.

An application of the proposed multiscale FE model onto the parametrized geometrical cytoskeleton model allowed the analyses of computed effective diffusion tensors and the resulting inaccessible area fractions within the cell. Interestingly, for various sampling domains within the cytoplasm the resulting homogenized diffusion tensors were almost perfectly isotropic, even if the filament density was artificially increased. Hence, the hypothesis whether the relation between the computed effective diffusion coefficients and the inaccessible area fraction can be approximated by means of a two-parameter power law, as suggested by Novak et al. [33] and shown in Eq (11), was tested to be statistically true for the experimentally observed obstacle geometries in eukaryotic cells.

The well-known exact representation of the effective tensor $\mathbf{D}^0$ is a series representation involving the geometric autocorrelation functions of all orders [67, 68]. In contrast, Eq (11) involves only the area fraction, which is the autocorrelation function of order zero. Comparing this with Golden and Papanicolaou [67], one could conjecture that Eq (11) is in fact an empirical closure that approximates high-order autocorrelation functions in terms of the average using the fitted parameters $\phi_c$ and $\mu$. A possible objection to this line of reasoning is that Golden and Papanicolaou [67] as well as Torquato [68] deal with a finite contrast case, meaning that the ratio between the diffusion coefficients of the two phases is finite. In the obstructed diffusion case, the contrast is infinite. Nevertheless, it is well known that for both finite and infinite contrasts, the volume fraction alone does not, in general, define $\mathbf{D}^0$ uniquely.

Otherwise, mixing two materials with prescribed volume fractions would always yield the same effective material properties. This contradicts a large body of literature on the effective property bounds, see, e.g., Milton [69]. However, Fig 5 emphasizes that Eq (11) yields a remarkable statistically accurate approximation of the relation between the relative diffusion coefficients and the inaccessible area fraction for the experimentally observed obstacle geometries in eukaryotic cells.

Once the parameters of the power law were estimated, it was possible to predict the effective diffusion tensor for a given inaccessible area fraction by means of Eq (11). In case of a direction-dependent growth and remodeling process of the cytoskeleton, the resulting effective diffusion tensor is expected to be anisotropic and can therefore not be predicted by the simple two-parameter power law accurately.

In general, validation of thermodynamical models by means of experimental data is a difficult task. In the context of the present study, Luby-Phelps et al. [70], e.g., published a relative diffusion coefficient of $D_{\text{cyto}}/D_{\text{aq}} = 0.167 \pm 0.009$ (mean ± standard error) for the diffusion of inert tracer particles (FTC-Ficoll) having a radius of 10.6 nm in the cytoplasm of Swiss 3T3 fibroblasts. Thereby, the diffusion coefficient for the cytoplasm $D_{\text{cyto}}$ is normalized by the aqueous diffusion coefficient $D_{\text{aq}}$. Similar values are presented in Popov and Poo [71] for the diffusional transport of globular proteins with a comparable hydrodynamic radius in the neurite cytoplasm. In the present study, relative effective diffusion coefficients $D_{11}^0/D_0$ and $D_{22}^0/D_0$ ranging between 1 and 0.9 were computed using the generated *in vitro* cytoskeleton model. At the first glance this discrepancy, however, is multicausal, including, e.g., the obvious fact that the present study only considers the actin filament network. Microtubules, intermediate filaments as well as different organelles that are usually found in eukaryotic cells were neglected. Hence, it can be concluded that the actin filament network itself is not able to slow down the diffusive transport of vesicles inside the cell by approximately 85%, as obtained in experimental studies. On the other hand, it is unclear to which extend the diffusion of nanoparticles in an aqueous solution, as used in Luby-Phelps et al. [70], differs from the diffusion of the same nanoparticles in pure cytosol, given that the cytosol consists of water and various biomolecular solutes.

Therefore, Luby-Phelps et al. [70] compared the diffusion of FTC-Ficoll fractions in cytoplasm with the diffusion in differently concentrated solutions of proteins, e.g., 10, 20 and 24% ovalbumin or 26% bovine serum. Computing the effective diffusion coefficient of the cytoplasm normalized by the diffusion coefficient of the concentrated protein solution (mimicking the cytosol), a relative value of approximately 0.83 is obtained for the 26% bovine serum albumin or 0.56 for the 24% ovalbumin solution. This, in fact, results in a much better agreement with the numerically obtained relative diffusion coefficients, where the homogenized diffusion coefficients $D_{11}^0$ and $D_{22}^0$ were normalized by the diffusion coefficient of the cytosol $D_0$. Furthermore, it emphasizes that concentrated protein solutions, such as the cytosol, exhibit a significantly higher effective viscosity than water. Nevertheless, the main focus of the present study was to investigate the role of specific actin filament density alterations onto the diffusive part of the vesicle-mediated cell transport.

Hereof, the numerical study on the impact of cytoskeletal alterations onto the vesicle transport inside an A549 cell finally revealed that a uniform four-fold increase of the amount of actin filaments together with a proportional increase of the network cross-linking proteins significantly slow down the diffusive vesicle transport. A hypothetical genetic defect or underexpression of filament network cross-linking proteins, such as Arp2/3 or filamin, results in an unbalanced proportion between the amount of filaments and cross-links, which ultimately yields a sparser and mechanically less stiff actin filament network. This, in turn, leads to a

faster vesicle motion due to the disruption of local diffusion barriers, as demonstrated by the numerical multiscale model for the diffusive vesicle transport. Even though, this numerical study was focused on the vesicle-mediated cell transport, the developed concepts and achieved experience summarized in this article are expected to be useful for the simulation of alternative intracellular diffusion processes which are necessary to ensure the optimal functionality of eukaryotic cells.

In summary, we introduced a very efficient numerical tool for modeling the diffusive vesicle transport inside a eukaryotic cell. We verified and validated the model by comparing the predicted effective diffusion coefficients with experimental data presented in Luby-Phelps et al. [70], and finally utilized the proposed multiscale method to analyze the influence of actin filament network alterations—as typically seen in cells experiencing increased or decreased mechanical loadings—onto the diffusion properties and the transport time of vesicles inside the cell. The presented model predictions in this study need to be validated in future experimental studies, e.g., where living cells are forced to increase their actin filament density by means of mechanical stimulation and the diffusion properties can be obtained using singe-vesicle tracking, as presented in Zhang et al. [13]. Note that the presented results only consider the diffusive part of the vesicle transport. Hence, the active transport along the microtubules needs to be chemically disabled in the experiments or cell without active transport need to be utilized, e.g., red blood cells.

According to Kaether et al. [72], the maximum vesicle velocity, experimentally quantified in cultured hippocampal neurons, is 5 $\mu$m/s, which is significantly faster than estimated in the current study. Hence, in order to model the vesicle dynamics more realistic, the consideration of the active transport via motor proteins along microtubules needs to be taken into account. This could, e.g., be realized by adjusting the proposed multiscale FE method such that the vesicle motion is modeled by means of a diffusion-advection problem instead of a pure diffusion problem. Furthermore, to make the model validation easier and to study the relationship between cytoskeletal alterations and a dysfunctional vesicle transport in more detail, the complete 3D structural information of the cytoskeleton needs to be considered in subsequent numerical analyzes. However, in the context of model validation via experimental studies, one major remaining limiting factor of current fluorescence microscopy imaging techniques is the fact that fluorescence-labeling of individual cell components directly affects the vesicle-mediated cell transport. Consequently, it is impossible to capture all the necessary structural information of the cytoskeleton and the vesicle dynamics at the same time. Finally, perhaps most importantly, this study has shown that the proposed multiscale FE model in combination with the approximation method of the effective properties on the basis of the inaccessible area or volume fraction seems to be a fast and powerful simulation tool to gain better insights into the microscopic effects of nanoscopic cytoskeletal alterations onto the vesicle transport within the whole cell.

## Supporting information

**S1 Data. Underlying numerical data for graphs.** Excel spreadsheet containing the underlying numerical data of Figs 1, 3, 5, 6, 7, 8 and 9.
(XLSX)

**S1 Appendix. Finite element implementation.** In this appendix, we provide some details on the implementation of a finite element method to find unique solutions of the diffusion problem and the cell problem, as introduced in Eqs (2) to (4).
(PDF)

**S2 Appendix. Validation and verification.** This appendix presents detailed descriptions and results of the validation and verification process regarding the introduced multiscale model for the diffusive vesicle transport based on the concept of homogenization.
(PDF)

## Acknowledgments

We thank Eleonore Fröhlich and Kristin Öhlinger from the Center for Medical Research at the Medical University of Graz, Austria, for sharing their image data.

## Author Contributions

**Conceptualization:** Daniel Ch. Haspinger, Sandra Klinge, Gerhard A. Holzapfel.

**Data curation:** Daniel Ch. Haspinger.

**Formal analysis:** Daniel Ch. Haspinger, Gerhard A. Holzapfel.

**Funding acquisition:** Sandra Klinge, Gerhard A. Holzapfel.

**Methodology:** Daniel Ch. Haspinger, Gerhard A. Holzapfel.

**Project administration:** Daniel Ch. Haspinger, Sandra Klinge, Gerhard A. Holzapfel.

**Resources:** Gerhard A. Holzapfel.

**Software:** Daniel Ch. Haspinger.

**Supervision:** Gerhard A. Holzapfel.

**Visualization:** Daniel Ch. Haspinger.

**Writing – original draft:** Daniel Ch. Haspinger.

**Writing – review & editing:** Daniel Ch. Haspinger, Sandra Klinge, Gerhard A. Holzapfel.

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
