## [Decision Letter · Decision Letter 0]

5 Oct 2020

Dear Dr. Holzapfel,

Thank you very much for submitting your manuscript "Numerical Analysis of the Impact of Cytoskeletal Actin Filament Density Alterations onto the Diffusive Vesicle-Mediated Cell Transport" for consideration at PLOS Computational Biology.

As with all papers reviewed by the journal, your manuscript was reviewed by members of the editorial board and by several independent reviewers. In light of the reviews (below this email), we would like to invite the resubmission of a significantly-revised version that takes into account the reviewers' comments.

We cannot make any decision about publication until we have seen the revised manuscript and your response to the reviewers' comments. Your revised manuscript is also likely to be sent to reviewers for further evaluation.

Sincerely,

Alison Marsden

Associate Editor

PLOS Computational Biology

Mark Alber

Deputy Editor

PLOS Computational Biology

Reviewer's Responses to Questions

**Comments to the Authors:**

Reviewer #1: The article presents a numerical method for investigating the effect of actin density on transport inside an eukaryotic cell. The model is verified using experimental data from the literature as well as analytical solutions for a simple one-dimensional problem. Further, a homogenization method was presented and validated using a Monte Carlo simulation. The advantages of the method presented involve precise control over the structure of the cell using only two parameters: the fractional area of inaccessibility and a fitting parameter. The FEM calculations were shown to be both fast and accurate. The authors found a drastic increase in time required for transport within a cell with increasing actin filament density consistent with literature. Overall, the reviewer believes that this paper should be accepted with major revisions. While the model is well-developed and justified, it does not present a clear argument consistent with its motivation.

1. The main critique of this article is the material presented as it relates to the argument and motivation of the paper. The introduction gives a good review of the importance of studying transport as well as an overview of previous studies, but only a small section of the article actually studies the phenomenon presented. Specifically, Section 3.4 presents the observed trends that the model predicts, but there is little discussion about the significance of these findings. This reviewer would suggest including a figure of the results and more discussion on the importance of investigating actin density in cell transport.

2. Although the model is effectively presented and validated, there is too much time dedicated to describing a homogenization method which is not used in the exposition. The second section of the paper is dedicated to presenting an FEM model that can be used to explore the role of actin density on cellular transport within the cytoplasm. As the diffusion tensor is known to be vary nonlinearly throughout the body and is a function of many variables, a homogenization method is presented. At each Gaussian point, a small sub-domain is created, discretized, and solved separately for the diffusion tensor. It is later used to validate a known power-law correlation presented by previous work from another group. For the actual study, this power law was used as it is less computationally expensive. Although the information is a compelling way to verify this power law, the reviewer finds that the presentation distracts from the message of the paper and adds unnecessary length. Furthermore, the power law was verified in previous papers against experiments, making further reducing the importance of this section.

3. There is not enough explanation of the benchmark cases studied and physical implications of the results. The figures do a good job of demonstrating a strong correlation between their model and the analytical solution of a one-dimensional diffusion problem, but a simple interpretation of the case studied is difficult due to the use of periodic boundary conditions. More time should be spent in motivating this benchmark problem to be relevant for the model that they are presenting.

In summary, more time should be spent discussing the importance of the results, including justification for the cases studied and figures that display their results. The presented model appears to be justified rigorously in its accuracy, but its usefulness is lost on the reader due to a lack of physical interpretation. The author acknowledges the simplification of the structure of the cytoplasm (the authors neglect the presence of organelles, etc. and only consider actin filaments as disrupting transportation), but they fail to present any compelling results to speak to the usefulness of this simplification. This reviewer believes that a more thoroughly explained study would help to justify the importance of the work being done. Finally, less time should be spend explaining a numerical method which is not used in the actual implementation.

Reviewer #2: In the manuscript entitled, 'Numerical analysis of the impact of cytoskeletal actin filament density alterations onto the diffusive vesicle-mediate cell transport', the authors present a mathematical model and a finite-element formulation for solving the partial differential equations resulting from the model.

There are several issues with the manuscript in its present form that make it hard to evaluate whether it achieves the goal of diffusion of vesicles as a function of actin filament density.

1) The introduction presents a very long summary of many topics in cell biology but doesn't actually outline the most recent literature on actin organization in cells or passive transport of vesicles. There is no mention of vesicle sizes or components. Furthermore, the authors have not consulted the recent advances in the literature on actin organization in cells (for example: https://www.nature.com/articles/ncb2044 and https://www.cell.com/structure/pdf/S0969-2126(16)30086-7.pdf). Indeed, a paper on actin organization without the mention of papers by stalwarts in the field such as Thomas Pollard is rather surprising. Furthermore, the actin organization is not the same across the different parts of the cells and actin is highly regulated by many actin binding proteins. Therefore, as written, it is not clear that the authors are actually modeling an actin network.

2) I read the paper by Miendl et al. 2017 on which the current paper is built on very carefully. In that study, they use nanoparticles and it is not clear that in the current study that nanoparticles are modeled. Nanoparticles and vesicles are used interchangeably (they are not). More importantly, the data in Miendl et al for actin is not at a resolution to reconstruct an actin network. Also, the data is only in 2D and therefore, it is not clear how the authors built a 3D meshwork of actin. Clearly the assumptions underlying these steps are important for building the 3D network. The resulting model does not represent the features of actin. A specific example of this is that the network constructed here has intersection angles of 46.39 degrees, while actin branch angles are known to be approximately 70 degrees (again many references that have established this point).

These two points essentially lead me to the conclusion that the biological assumptions underlying this work are flawed.

3) The model that the authors develop for the vesicle concentration is a diffusion model. I have the following issues with the model.

3a) What does vesicle concentration mean? Number of vesicles per unit volume? are vesicles present in a large enough density that a concentration can be defined for them? In a cube of 1 cubic micron, how many such vesicles are there?

3b) What do the boundary conditions translate to in the physical domain? Are the boundaries representing the plasma membrane? The organelle membrane? Or the filaments? Do the filaments provide a physical barrier for diffusion?

3c) It is also noteworthy that the authors do not reference the works by Paul Bressloff and other colleagues who have developed extensive models on diffusion of particles and vesicles in crowded spaces.

3d) Does the vesicle size not matter? There is no consideration given to this factor. What about thermal fluctuations in the cellular environment?

It appears that the authors developed a mathematical and finite element model for a diffusive process, which in and of itself is fine. However, this model has very little to do with the biological process the authors claim to study. If the authors wish to write a paper on the mathematical aspects of their model, then they should do so and consider sending it to a suitable journal. If the authors wish to address a major problem in cell biology, then the details of the biological process are important and more than one set of experiments should be taken into account. Furthermore, the model should be validated against data that is already published and then generate experimentally testable predictions. Right now, this manuscript does neither.

**Have all data underlying the figures and results presented in the manuscript been provided?**

Reviewer #1: Yes

Reviewer #2: Yes

PLOS authors have the option to publish the peer review history of their article (what does this mean?). If published, this will include your full peer review and any attached files.

Reviewer #1: No

Reviewer #2: No
---

## [Decision Letter · Decision Letter 1]

9 Feb 2021

Dear Dr. Holzapfel,

We are pleased to inform you that your manuscript 'Numerical Analysis of the Impact of Cytoskeletal Actin Filament Density Alterations onto the Diffusive Vesicle-Mediated Cell Transport' has been provisionally accepted for publication in PLOS Computational Biology.

Best regards,

Alison Marsden

Associate Editor

PLOS Computational Biology

Mark Alber

Deputy Editor

PLOS Computational Biology

Reviewer's Responses to Questions

**Comments to the Authors:**

Reviewer #1: In the revised edition of their manuscript, the authors have presented a much more compelling argument for the usefulness and relevance of their numerical model and analysis. In particular, the details of their FEM implementation and validation have been moved to the appendices, at great benefit to the readability of the manuscript. Indeed, the significance of the multi-scale implementation was lost on me the first time around as it was bogged down by seemingly irrelevant mathematical analysis. I quite enjoyed reading this revised manuscript and find it to have a refreshingly clearer story than the original.

I like the added discussion and analysis of the impact of vesicle radius onto the inaccessible area fraction in the sampling domain; the addition of figure 6 aids in promoting the physically motivated parameters and capabilities of their model. Further, the connection to their simulations and the approximated diffusive coefficients of Novak et al. is much stronger, and I am pleased to now appreciate what is clearly a well-formulated and implemented homogenization technique with a clear connection to experimental data.

A better understanding of the model certainly lent to a more convincing article overall. The parametric study (Table 2/Figures 7/9) is certainly interesting, albeit a bit difficult to navigate. Nevertheless, the combination of Table 2 with Figure 7 aids in understanding the cases studied and their physical implications on the system. I could suggest a minor revision or two to further aid in the ease of understanding (for instance, a visual indication on Figure 7 of the parameter that remains constant in each row), and I believe it would make more sense for Figure 8 to go before Figure 7 so that the two figures relating directly to the parametric study are sequential, but this perhaps speaks more to my own preference than anything.

Finally, I do maintain that the discussion and validation sections are a bit dense. The topic is clearly well-researched, but the discussion of the works of Golden and Papanicolaou (1983) and Luby-Phelps et al. (1987), for instance, come across as afterthoughts in the final section of the paper. If the authors wish to include these points of discussion, I would opt to include a more concise addition of these citations in relevant sections of the main article. Further, if the appendices are indeed restricted to a simple demonstration and validation of their implementation, I do not believe that four additional figures are needed to illustrate this. These last two points are perhaps more closely aligned with my own preferences as well, and if the authors insist on their chosen style of presentation, I do not believe that it greatly hinders the quality of the manuscript.

In summary, the revised edition of this manuscript is a great improvement of the previous one. I thoroughly enjoyed reading their analysis through a clearer lens, and although a few changes could be made to improve the article to my own preferences, I believe that it would make an impactful publication as-is.

Reviewer #2: The authors have attempted to address my previous comments and clearly put in a substantial amount of effort. I believe that the manuscript is easier to follow than before. However, I still have the following issues.

1) Biological Premise: The fundamental premise of large vesicles freely diffusing in cells is not really biologically relevant. Vesicles are transported by motor proteins. In response to my previous comment in the summary (However, the model has little to do with the biological process ...), the authors responded that they present an efficient numerical multiscale FE tool for modeling the diffusive part of vesicle-mediated(sic). I'm questioning the relevance of assuming that vesicles diffuse through the cytosplasm. I did a literature search to see what fraction of vesicles diffuse freely in the cytoplasm and could find mention vesicle diffusion in the proximity of the plasma membrane prior to scission or fusion (see e.g.https://www.ncbi.nlm.nih.gov/pmc/articles/PMC2599850/). Almost all other readings point toward active transport (https://www.nature.com/articles/srep04481 and references therein for example).

To alleviate this concern, let's assume that the authors are modeling passive diffusion of large macromolecular aggregates through the cytoplasm. This would make more sense in the context of the model that they have developed.

2) Text: The text is really verbose and really doesn't get to the point at all. In fact, the introduction and discussion both meander without getting to the point  to account for diffusion in crowded spaces such as the cytosol, we have developed an efficient finite element method.

3) Predictive capability: From the results regarding effective diffusion coefficient (Figure 5) and Figure 6 can the authors draw any predictions that can be compared against experiments? (just one example: https://www.ncbi.nlm.nih.gov/pmc/articles/PMC2173574/)

4) D_{in vitro} calculation: why use the viscosity of water when the viscosity of the cytoplasm is known? https://pubs.acs.org/doi/10.1021/acs.jpclett.0c01748 for example (there are many others)

Again, I maintain that the authors have a strong computational method for diffusive transport in crowded spaces; its relevance to biology as applied in the manuscript is less obvious since the authors seemed to have missed some of the basics of vesicle transport.

**Have all data underlying the figures and results presented in the manuscript been provided?**

Reviewer #1: None

Reviewer #2: Yes

PLOS authors have the option to publish the peer review history of their article (what does this mean?). If published, this will include your full peer review and any attached files.

Reviewer #1: No

Reviewer #2: No

---

## [Editor Report · Acceptance letter]

21 Apr 2021

PCOMPBIOL-D-20-01624R1 

Numerical Analysis of the Impact of Cytoskeletal Actin Filament Density Alterations onto the Diffusive Vesicle-Mediated Cell Transport

Dear Dr Holzapfel,

I am pleased to inform you that your manuscript has been formally accepted for publication in PLOS Computational Biology. Your manuscript is now with our production department and you will be notified of the publication date in due course.

With kind regards,

Katalin Szabo
